# SARS-CoV-2 Infection and Preeclampsia—How an Infection Can Help Us to Know More about an Obstetric Condition

**DOI:** 10.3390/v15071564

**Published:** 2023-07-17

**Authors:** Otilia González-Vanegas, Oscar Martinez-Perez

**Affiliations:** 1Obstetric Department, Hospital San Cecilio, Granada 1, 18016 Granada, Spain; otigonzalez16@gmail.com; 2Obstetric Departament, Hospital Universitario Puerta de Hierro 2, 28222 Majadahonda, Spain

**Keywords:** SARS-CoV-2, COVID-19, preeclampsia, pregnancy, inflammation, myocardial injury

## Abstract

Pregnant women with SARS-CoV-2 infection have a significantly higher risk of maternal death, ICU admission, preterm delivery, and stillbirth compared to those without infection. Additionally, the risk of preeclampsia (PE) increases in pregnant women infected with SARS-CoV-2, particularly in severe cases. The association between COVID-19 and PE is likely attributed to various mechanisms, including direct effects of the virus on trophoblast function and the arterial wall, exaggerated inflammatory response in pregnant women, local inflammation leading to placental ischemia, SARS-CoV-2-related myocardial injury, cytokine storm, and thrombotic microangiopathy. This paper aims to explore the similarities between PE and SARS-CoV-2 infection, considering COVID-19 as a valuable study model. By examining these parallels, we can enhance our knowledge and comprehension of PE. We wish to emphasize the potential for COVID-19-induced myocardial injury in pregnant women and its connection to the increased maternal mortality rate.

## 1. Introduction

The World Health Organization declared the outbreak of severe acute respiratory syndrome (SARS) caused by coronavirus 2 (CoV-2) as a pandemic on 11 March 2020 [1]. Subsequently, a multitude of studies have been published investigating the disease’s progression, the impact of different virus strains, and the transformation into a milder respiratory illness due to widespread vaccination.

According to the most recent data, as of June 2023, there have been over 768 million confirmed cases of coronavirus disease-19 (COVID-19) worldwide. Additionally, the disease has claimed the lives of more than 6.9 million individuals, while approximately 65.67% of the global population has received a complete primary series of the vaccine. Notably, more than 13.3 billion vaccine doses have been administered [2].

COVID-19 typically manifests as fever, dry cough, and fatigue. However, approximately 14% of cases can progress to severe pneumonia, and 5% can develop into SARS, necessitating admission to the intensive care unit (ICU) for specialized respiratory support. It is worth noting that, although COVID-19 primarily affects the respiratory system, it can also have significant systemic effects, such as hypertension, kidney disease, thrombocytopenia, and liver injury [3].

Numerous studies have been conducted to assess the implications of SARS-CoV-2 infection in pregnant women for multiple reasons. The susceptibility of pregnant women to the virus was a concern, given the lack of prior research on the impact of other coronaviruses on fetal health during pandemics. Providing guidance to healthcare professionals and expectant mothers regarding the potential consequences of viral infection on maternal and fetal well-being was of utmost importance [4,5].

During pregnancy, the risk of adverse obstetric and neonatal outcomes is heightened by various respiratory viral infections [6]. The physiological and immunological changes inherent to a normal pregnancy can result in systemic effects that make pregnant individuals more susceptible to complications from respiratory infections. Alterations in maternal cardiovascular and respiratory systems, including increased heart rate and oxygen consumption, higher stroke incidence, reduced lung capacity, and immune adaptations to accommodate the tolerance of the fetus to different antigenic characteristics, collectively contribute to an increased likelihood of pregnant women experiencing severe respiratory illnesses.

This association has also been observed in previous cases where pregnant women contracted two other pathogenic coronavirus infections: SARS and Middle East respiratory syndrome [7]. Influenza studies have consistently shown an elevated risk of maternal morbidity and mortality in comparison to non-pregnant women when pregnant women acquired respiratory viral infections [8].

In addition to the respiratory symptoms associated with COVID-19, the presence of systemic effects has been increasingly recognized. Several authors have observed a potential association between SARS-CoV-2 infection during pregnancy and an elevated relative risk of preeclampsia (PE) and other pregnancy complications [9,10,11,12,13]. Indeed, pregnancy has been classified as a comorbidity in the CDC’s list of risk factors for severe COVID-19, highlighting the heightened vulnerability of pregnant individuals to experience severe forms of the infection and related pregnancy complications [14].

PE is a condition that complicates pregnancy, poses a substantial risk to the health of both the mother and the fetus, resulting in adverse perinatal outcomes and induced premature birth. PE occurs in approximately 2–4% of pregnancies, and its precise etiology remains incompletely understood. However, it is believed to involve maternal vascular malperfusion and cardiovascular maladaptation due to an imbalance in angiogenic factors, endothelial dysfunction, coagulopathy, and insufficient complement regulation [15]. Acknowledging the syndrome-based nature of PE, the American College of Obstetricians and Gynecologists’ Task Force on Hypertension in Pregnancy made a significant change in 2013 with updates and recommendations made in 2019 and 2020. They eliminated the requirement of proteinuria for diagnosis and instead focused on new-onset hypertension accompanied by indicators of organ dysfunction. These indicators include thrombocytopenia, hypertransaminasemia, elevated serum creatinine in the absence of other kidney disease, pulmonary edema, or the onset of cerebral or visual disturbances [16].

The clinical and laboratory manifestations of PE can differ based on the timing of onset during pregnancy, whether it occurs early or late in gestation [15]. There are notable similarities between the clinical symptoms observed in patients with PE and those with severe COVID-19 disease [10].

The potential link between SARS-CoV-2 infection and the risk of PE has prompted extensive research, including meta-analyses, systematic reviews, cohort studies, and case reports. This area of study is regarded as a public health priority, calling for specialized clinical management to minimize adverse outcomes in pregnancy [9,10,17]. However, it is important to acknowledge that certain publications on the course of COVID-19 in pregnancy have found no significant association with PE [17].

## 2. Discussion

### 2.1. How COVID-19 and PE

#### 2.1.1. Regarding the Presence or Absence of Symptoms of COVID-19

A meta-analysis by Conde Agudelo investigated the connection between SARS-CoV-2 infection during pregnancy and the occurrence of PE. The study included 15,524 pregnant women infected with the virus. The findings showed that SARS-CoV-2 infection increased the risk of PE by 62% compared to pregnant women without the infection. This increased risk was observed in both symptomatic and asymptomatic cases, but it was more common in symptomatic individuals. Even after adjusting for other risk factors, the association between COVID-19 and PE remained significant. The study concludes that pre-existing risk factors alone cannot fully explain the relationship, providing strong evidence that pregnant women with SARS-CoV-2 infection are at a higher risk of developing PE [9]. Table 1

Prior to the availability of vaccines, previous studies have indicated that pregnant women with symptomatic COVID-19 had worse outcomes compared to those without symptoms, especially in cases with severe symptoms. These severe cases were found to have a higher risk of developing PE when compared to asymptomatic patients [12].

Regarding the same issue in a prospective multicenter study conducted by Álvarez Bartolomé and colleagues, a cohort of 1347 pregnant women infected with SARS-CoV-2 during the first wave of the pandemic was recruited from 78 centers in Spain. The objective of the study was to investigate the causes and risk factors for ICU admission. The incidence of ICU admission in this cohort was 2.6% prior to delivery. Non-obstetric factors were the main causes of ICU admission, with 40% attributed to worsening maternal condition and respiratory failure due to SARS-CoV-2 pneumonia. Furthermore, 31.4% of admissions were a result of a combination of COVID-19 symptoms and obstetrical complications, particularly PE (37.1% vs. 4.3% in pregnant women not admitted to the ICU) and hemorrhagic events (such as postpartum hemorrhages and/or abruptio placentae) (20% in ICU admissions vs. 4.8%). However, the authors emphasized that the risk of ICU admission was significantly higher only when PE or obstetrical hemorrhagic events were associated with symptoms of COVID-19, including severe pneumonia [18].

Similarly, the INTERCOVID group conducted a multinational cohort study involving patients from 43 institutions in 18 countries. Their objective was to investigate the relationship between COVID-19 symptoms and the risk of maternal morbidity and mortality, specifically focusing on PE. The results of the study indicated that pregnant women with COVID-19 symptoms, particularly those experiencing respiratory symptoms lasting several days, had a higher risk of developing PE compared to asymptomatic patients. Although specific statistical results are not necessary for comprehension, the study found an increased risk of PE in symptomatic patients compared to asymptomatic patients [11,19].

COVID-19 and PE are not the only conditions that share similarities. Prior to the pandemic, several disorders were identified that mimic PE due to similar clinical and laboratory findings. These disorders also have common underlying causes in their pathophysiology, including endothelial cell dysfunction, platelet activation, microvascular thrombosis, vasospasm, and reduced tissue perfusion. Some examples of these disorders include gestational hypertension, acute fatty liver during pregnancy, hemolytic uremic syndrome, chronic kidney disease, acute exacerbation of systemic lupus erythematosus, thrombotic thrombocytopenic purpura, severe hypothyroidism, and sepsis. These similarities indicate that, although there is a shared basis between these two conditions, it is important to differentiate them to ensure proper treatment and prevent complications for both the pregnant woman and her baby. In line with this, a study conducted by Mendoza et al. and recently Serrano et al. have described a syndrome in pregnant women with COVID-19 that bears similarities to pre-eclampsia (referred to as a pre-eclampsia-like syndrome). These findings emphasize the importance of not resorting to premature delivery in these patients, as it could potentially worsen neonatal outcomes when treating them as if they had PE [20,21,22].

Perinatal outcomes that occur when both diseases are associated have also been examined; in a multicenter prospective observational study conducted in Spain, researchers analyzed a cohort of pregnant women who tested positive for SARS-CoV-2 and compared them to a group of women who tested negative. The study aimed to investigate the relationship between maternal infection and perinatal outcomes. The results showed that the group of women with SARS-CoV-2 infection had higher rates of premature rupture of membranes (PROM), venous thrombotic events, and a significantly higher incidence of severe PE compared to the group without infection. However, the researchers suggested that the incidence of PE in the infected cohort may have been overestimated due to the potential misinterpretation of COVID-19-related markers as signs of PE, given the presence of inflammation in both conditions [23].

#### 2.1.2. Considering the Severity of COVID-19

The severity of the COVID-19 has also been associated with a higher incidence of PE, as evidenced by several studies. Jonathan Lai et al. conducted a retrospective observational study to investigate the association between the severity of SARS-CoV-2 infection and the likelihood of developing PE. The pregnant women with COVID-19 and PE were categorized into four groups based on the severity of COVID-19: asymptomatic, mild, moderate, and severe. The risks obtained were 1.9%, 2.2%, 5.7%, and 11.1% respectively. The main findings revealed a dose−response relationship between the severity of SARS-CoV-2 infection and the risk of developing PE, with the frequency of PE increasing as the severity of COVID-19 increased.. Additionally, they also reported a median interval of 16 days (interquartile range, 7–61 days) between the onset of SARS-CoV-2 infection and the diagnosis of PE, further strengthening the existence of an association between the two conditions [24].

A significant finding in the meta-analysis conducted by C Agudelo was the presence of a dose−response relationship between the severity of SARS-CoV-2 infection and the increased risk of developing PE and preterm birth. This suggests that severe SARS-CoV-2 infection indirectly influences pregnancy by initiating a vascular condition commonly observed during gestation, such as PE, which consequently impacts the patient’s prognosis [9].

#### 2.1.3. Comparing the Incidence of PE in Relation to Gestational Age at Infection

The timing of infection during pregnancy played a crucial role in determining the occurrence of unfavorable outcomes for both the mother and the baby. Only a limited number of studies have categorized the negative outcomes of patients based on the specific stage of pregnancy when the infection took place.

Badr et al. conducted a study aimed to compare maternal and fetal outcomes based on the timing of infection during pregnancy. The investigators found that adverse obstetric outcomes were more severe when the infection occurred after 20 weeks of gestation, and perinatal outcomes were unfavorable when the infection occurred after 26 weeks. As a result, the authors suggested that vaccination should be administered promptly after pregnancy diagnosis to mitigate these risks [25].

Rosenbloom conducted an analysis to determine the time interval between the onset of COVID-19 and PE and found that the association between the two conditions was weaker when the infection occurred after 32 weeks of pregnancy compared to before 32 weeks (OR 2.88 vs. 2.74). This association was statistically significant only prior to the 32nd week of pregnancy. Furthermore, Rosenbloom reported that it took approximately 3.79 weeks from SARS-CoV-2 infection to the onset of PE, suggesting that in full-term pregnancies, there was insufficient time for the development of PE following SARS-CoV-2 infection. However, other studies indicate that the risk of PE remains regardless of the gestational age at the time of SARS-CoV-2 infection [26]. 

In summary, there is sufficient evidence supporting the increased incidence of PE in pregnant women with COVID-19, regardless of the presence of pre-existing comorbidities. The onset of PE is directly related to the presence and severity of respiratory symptoms due to COVID-19, particularly when the infection occurs before the 32nd week of gestation.

**Table 1 viruses-15-01564-t001:** Authors reported preeclampsia (PE) rate, and/or OR of PE.

First Author	Year of Publication	Country	Study Desing	Setting	N	Poblation	Rate of PE (%)	OR PE (IC 95%)
Di Mascio [7]	2020	UK	SR-M	19 studies	79		16.2% (4.2–34.1)	
C Agudelo [9]	2021	USA	SR-M	28 OS	790,954 15,524 (+)	SARS +	Infected 7% Non infected 4.8%	1.62 (1.45–1.82) * Symptoms 2.11 (1.59–2.81) No symptoms 1.59 (1.21–2.10)
Shu Qin Wei [13]	2021	China	SRM	42 OS	438,548	SARS +		Mild 1.33 (1.03–1.73) Severe 4.16 (1.51–11.15)
J. Juan [17]	2020	China	SR	24 studies	324	SARS +	1.7%	
Papageorghiou [11]	2021	INTERCOVID	POS	43 Hospitals 18 Countries	2184	SARS +/− PE +/−	Infected 8.1% Non infected 4.4%	RR all 1.77 (1.25–2.52) Nulliparous 1.89 (1.17–3.05) * Parous 1.64 (0.99–2.72)
Villar [19]	2021	UK Multinational	PCS	43 Hospitals 18 Countries	2130	SARS +/−		[RR] 1.76 (1.27–2.43) Asymptomatic 1.63 (1.01–2.63)
Mendoza [20]	2020	Spain	POS	Terciary Hospital	42	SARS + >20 w	11.9%	
Cruz Melguiso [23]	2020	Spain	PCS	78 Spanish Hospitals	2754	SARS +/−	Infected 40.6% * Non infected 15.6%	3.69 (1.62–8.39) *
Lai J. [24]	2021	UK	ROS	14 Health Services	1223	SARS +	No symptoms 1.9% * Mild COVID-19 2.2% * Moderate 5.7% * severe 11.1% *	
Badr DA [25]	2021	Belgium Multinational	RCS	4 European University Hospitals	10,925	SARS +/−	Exposed 2.44% vs. Unexposed 1.89% *	
Rosembloom [26]	2021	USA	RCS	Terciary Hospitals	249	SARS+/−	Infected 13.3% Non infected 9.4%	
Birol [27]	2022	Turkey/UK	RCS	2 Terciary Hospitals	1286 Unvaccinated	SARS +		Delta/Pre-Delta 0.66 (0.27–1.61) Omicron/Pre-Delta 1.851(0.65–5.08)
Jie Deng [28]	2022	China	SR-M	18 studies	133,058	SARS +	PP Wild Type 1.06% (0.53–1.59) * Pre Delta 23% (16.99–29.03) * Delta 9.63% (1.47–17.80) * Omicron 12.52% (−10.3–35.33)	

POS: prospective observational study, SR: systematic review, SR-M: systematic review and metanalysis, ROS: retrospective observational study, RCS: retrospective cohort study, PCS: prospective cohort study PP: Pooled Prevalence, OS: observational study. * *p* < 0.05

### 2.2. Can the Association between COVID-19 and PE Be Explained?

The association between PE and COVID-19 is clear, although it remains uncertain whether this association is causative in both directions. Further investigation is needed to understand the underlying mechanism in order to differentiate and implement targeted therapeutic interventions. Several theories have been proposed to explain the association between COVID-19 and PE. It is likely that this association is not due to a single mechanism, but rather a combination of several factors. These include:(I)Direct effects of the virus on trophoblastic function and the arterial wall, which can result in endothelial damage and dysfunction.(II)Acute atherosis, a specific lesion observed in the spiral arteries similar to atherosclerotic lesions in the coronary arteries [29].(III)Local inflammation leading to placental ischemia.(IV)Indirect effects due to exaggerated inflammatory responses in pregnant women, including the release of cytokines such as IL-6.(V)Thrombotic microangiopathy (TMA).(VI)Imbalance between pro-angiogenic and anti-angiogenic factors.(VII)SARS-CoV-2-related myocardial injury, as suggested by recent and ongoing research.

These mechanisms may interact and contribute to the development of PE in pregnant women with COVID-19.

#### 2.2.1. The Door Is the Clue

The SARS-CoV-2 virus gains entry into host cells by attaching to the membrane receptors angiotensin-converting enzyme 2 (ACE2) and CD147. Once attached, the viral spike protein (S) undergoes cleavage with the help of host proteases, allowing it to merge with the cell membrane. The key host protease responsible for initiating this process and facilitating viral entry is TMPRSS2, a type II transmembrane serine protease [30,31]. In addition to ACE2, SARS-CoV-2 impacts other elements of the renin-angiotensin system (RAS), including ADAM17, which primarily controls the levels of ACE2 on the cell membrane and consequently reduces its expression [32]. ACE2, a membrane-bound exopeptidase primarily found in the kidney, lungs, and heart, is a critical component of the RAS, which regulates the conversion of angiotensin II (Ang II) to angiotensin 1–7 (Ang 1–7) and the conversion of angiotensina I (Ang I) to angiotensina 1–9 (Ang 1–9). Upon binding of SARS-CoV-2 to ACE2 receptors, the RAS system is downregulated, resulting in reduced levels of vasodilatory Ang 1–7, which is no longer counteracted by the pro-inflammatory and vasoconstrictive effects of Ang II.

In pregnancy, the RAS system plays a crucial role in placental function, controlling trophoblast proliferation, angiogenesis, and blood flow. It significantly influences utero-placental blood flow through the modulation of vasoconstrictor and vasodilator pathways [30,32].

These changes in the RAS system strongly support a similar origin in the pathophysiology of PE and COVID-19 disease [33]. The susceptibility to viral infection during pregnancy can vary; the expression of ACE2 in cells of different tissues is dependent on gestational age and is higher during the first trimester [31,34].

During normal pregnancy, there is a rise in plasma Ang-(1–7) levels and a decrease in ACE concentration. However, in women with PE, this pattern is reversed. Women who give birth to small-for-gestational-age babies also exhibit higher ACE concentrations. Placental cells show elevated levels of ACE2, and TMPRSS2 increases their affinity to the spike protein [31].

Histopathological studies conducted on patients with COVID-19 have revealed an increase in the expression of ACE2 protein in lung tissues, in contrast to non-infected patients. This abnormal regulation of ACE2 levels serves as a significant shared biomarker for both PE and COVID-19 [32].

Based on these findings regarding the RAS system, even though each pathology accesses it through different pathways, they exhibit a similar pattern, resulting in overlapping clinical manifestations.

#### 2.2.2. Interleukins as Possible Explanation

Investigating the underlying mechanisms and potential therapeutic targets for maternal conditions, particularly PE, has gained significant research attention, especially in light of the COVID-19 pandemic and its association with an increased incidence of PE. Understanding the impact of inflammation on pregnancy-related disorders has become a crucial area of study in this context.

COVID-19 and PE are characterized by an associated proinflammatory state known as cytokine storm, which acts as a risk factor for a more severe disease progression. In both cases, the levels of proinflammatory enzymes such as IL-6, tumor necrosis factor (TNF)-alpha, and serum ferritin are elevated. The proinflammatory state observed in COVID-19 may contribute to hypoxic damage in the placenta, leading to the development of PE. Additionally, it may also increase the likelihood of intrauterine growth retardation and preterm delivery [35].

IL-6 is one of the cytokines that has been extensively studied in both human parturition and adverse pregnancy conditions. It is sometimes referred to as a causal factor for adverse outcomes, including infections and inflammation associated with preterm birth and preterm PROM [36]. IL-6 plays a role in regulating acute and chronic inflammatory responses in autoimmune disorders and endothelial cell dysfunction. It also serves as an independent risk factor for cardiovascular diseases [37].

SARS-CoV-2 infection primarily spreads through the upper respiratory tract, initiating immune system stimulation, and progresses to the lower respiratory tract, where it replicates and triggers various alarm signals. As mentioned previously, the viral S glycoprotein acts as the “key” to enter cells by binding to the ACE-2 receptor expressed in pulmonary alveolar epithelial cells, endothelium, and alveolar macrophages [30,37,38].

Activation of alveolar immune effector cells leads to the release of proinflammatory cytokines, including IFNα, IFNγ, IL-1β, IL-6, IL-12, IL-18, IL-33, TNFα, TGFβ, as well as chemokines such as CXCL10, CXCL8, CXCL9, CCL2, CCL3, CCL5. Excessive immune system response to SARS-CoV-2 infection results in respiratory and multiorgan complications. IL-6, when produced uncontrollably, promotes pro-inflammatory activity and contributes to the pathogenesis of cytokine release syndrome [37,38].

In the pathogenesis of PE, a similar mechanism has been described, where placental stress triggers an aberrant systemic inflammatory response that can lead to maternal and neonatal complications. Maternal levels of inflammatory mediators, such as IL-6, IL-8, TNF, intercellular adhesion molecule-1, vascular cell adhesion molecule-1, and P-selectin, are elevated in PE [39].

These findings provide support for the idea of an enhanced systemic inflammatory response in PE, which is believed to contribute significantly to vascular dysfunction. Hallmarks of this inflammatory response in the pregnancy disorder include endothelial dysfunction, persistent activation of leukocytes and platelets, and elevated levels of inflammatory cytokines. The increased inflammatory response conducts to oxidative stress and vasoconstriction, resulting in multiorgan involvement that is shared by both conditions [40,41].

Finally, laboratory studies have shown that IL-6 does not pass through the placenta; considering the low transplacental transmission of SARS-CoV-2, it can be concluded that the placental barrier effectively defends against the consequences of the COVID-19 cytokine storm in the fetus [39].

#### 2.2.3. TMA, COVID-19, and PE

In coagulopathy, clot formation occurs through the activation of both the coagulation and fibrinolytic cascades. In the case of COVID-19, which is likely triggered by viral sepsis, coagulation activation leads to the depletion of clotting factors. This condition can manifest as either thrombotic or hemorrhagic events. However, when coagulopathy is associated with pregnancy, the process becomes more complex due to the physiologically hypercoagulable state of pregnancy, which involves an increase in coagulation factors like fibrinogen and D-dimers [42].

TMA encompasses a group of diverse conditions characterized by thrombocytopenia, microangiopathic hemolytic anemia, and multiple organ dysfunction syndromes. These conditions are associated with high rates of morbidity and mortality. Endothelial damage and micro thrombosis play a key role in the pathogenesis, leading to platelet consumption, destruction of red blood cells by fibrin filaments, and ischemia in organs and tissues due to vascular occlusion. Disseminated intravascular coagulation often occurs concurrently [43]. TMA can be classified into primary and secondary forms. Primary TMA includes hemolytic uremic syndrome with predominant renal involvement and thrombotic thrombocytopenic purpura with multiple organ dysfunction syndrome. Secondary TMA is associated with various systemic conditions such as pregnancy, infections, severe hypertension, connective tissue diseases, cancer, autoimmune diseases, organ transplantation, and certain medications [43].

A significant number of complications during pregnancy and the postpartum period can lead to the development of TMA. Particularly dangerous are conditions such as PE, eclampsia, and HELLP syndrome. Pregnancy can trigger primary TMA in individuals with an existing predisposition [42].

Viral infections have been extensively documented to have a significant role in the development of TMA. The virus can directly harm endothelial cells through various mechanisms, such as inducing the expression of adhesion molecules, releasing von Willebrand factor, promoting platelet adhesion, or activating alternative pathways like complement, which ultimately leads to TMA. Although the exact mechanism of viral microangiopathy remains unclear, it is evident that direct endothelial wall damage plays a crucial role [44].

Emerging evidence suggests that hospitalized COVID-19 patients experience a hypercoagulable state. Considering that pregnancy itself leads to a hypercoagulable shift, it is reasonable to suggest that COVID-19 infection during pregnancy poses a high risk of maternal thrombotic complications. This risk is particularly elevated in pregnant women with antiphospholipid antibodies, secondary infections, TMA, sepsis, comorbidities, and severe obstetric complications. Inflammation also contributes to a hypercoagulable state through endothelial cell damage, leading to thrombosis. Elevated levels of fibrinogen, triggered by high IL-6 release (cytokine storm), contribute to the formation of fibrin clots, resulting in the formation of unwanted microthrombi [45].

Complement deposits have been observed in the lungs and skin of COVID-19 patients, suggesting another potential pathway for the activation of thrombotic events. Histological analysis of these patients revealed the presence of complement complexes such as C5b-9, C4d, and MASP2 in the micro vessels of the lung and skin. The formation of these complement complexes triggers the activation of the coagulation system, leading to the formation of fibrin blood clots. Concurrently, there was a significant elevation in the blood levels of D-dimer [45].

In summary, the literature emphasizes the significant role of thrombotic complications in the morbidity and mortality of severely affected patients, including pregnant women, during the COVID-19 pandemic. Therefore, it is recommended to consider the use of prophylactic or therapeutic doses of low-molecular-weight heparin in all patients, unless contraindicated, to mitigate these risks [46]. However, it should be noted that in certain cases, this treatment may not effectively prevent thrombotic complications, indicating the involvement of additional mechanisms, such as TMA.

#### 2.2.4. Imbalance between Pro-Angiogenic and Anti-Angiogenic Factors. The Usefulness of Angiogenic Markers in the Differential Diagnosis of PE

SARS-CoV-2 affects the secretion of cytokines and growth factors [47]. The importance of biomarkers in diagnosing PE is becoming recognized. Literature reviews have identified several biomarkers that demonstrate enough specificity and sensitivity to be considered as potential markers. An effective biomarker should have the ability to detect PE at an earlier stage of the disease. Additionally, considerable research has been conducted on different prediction systems for PE [48].

Placental dysfunction is at the core of various perinatal conditions, such as PE and FGR. Several angiogenic factors are involved in placental dysfunction. VEGF-A (vascular endothelial grow factor A), an essential factor for placental vascular development, affects vascular permeability and proliferation and migration of endothelial cells. Placental growth factor (PlGF), another member of the proangiogenic VEGF family, is highly expressed in the placenta and enhances the effects of VEGF-A. On the other hand, sFlt-1 (soluble fms-like tyrosine kinase 1), an antiangiogenic member of the VEGF family, plays a crucial role in maintaining angiogenic homeostasis during pregnancy. Both sFlt-1 and PlGF are present in the placenta, vascular endothelial cells, osteoblasts, smooth muscle cells, fibroblasts, and monocytes [49].

Abnormal serum levels of the angiogenic factors sFlt-1 and PlGF indicate placental dysfunction and may show changes weeks before the onset of a pregnancy complication. In women with suspected PE, a normal sFlt-1/PlGF ratio can reliably predict the absence of short-term development of PE [50].

The utility of the Flt-1/PlGF ratio as a diagnostic tool, predictive marker, and monitoring tool for PE has been well established by numerous studies prior to the pandemic [50]. During the pandemic, this diagnostic laboratory tool was used in pregnant women infected with SARS-CoV-2 and showing clinical signs of PE, aiming to distinguish between the two conditions. Some studies found altered levels of these markers in patients with severe COVID-19 and symptoms of PE, conversely, in others, there was no clear correlation between the changes in these values and the occurrence or severity of PE. The author suggests considering the limited sample size in many research studies and the timing of blood sample collection, which was primarily done at the time of COVID-19 diagnosis rather than at the diagnosis of PE. This difference in timing could introduce variability in the analysis [51,52].

In PE, the levels of both sFlt1 and Ang II are elevated, while PlGF is significantly reduced. On the other hand, COVID-19 patients exhibit decreased PlGF levels but normal sFlt1 levels, with minimal changes in the sFlt1/PlGF ratio. The alterations in sFlt-1 and PlGF levels vary depending on the timing of PE diagnosis, with early-onset PE showing more pronounced changes compared to late-onset PE [53].

Regarding the biomarkers in hypertensive disorders of pregnancy (HDP) Soldavini et al. conducted a study and found that non-COVID-19 patients with HDP had significantly higher sFlt-1/PlGF ratios compared to both hypertensive and normotensive COVID-19 patients. This suggests that there are independent pathways of inflammation and angiogenic balance in HDP cases, regardless of their COVID-19 status. The study also showed that placental biomarkers were not correlated with the severity of symptoms, except in cases of severe respiratory failure [48].

Furthermore, Espino-y-Sosa conducted a study and found that pregnant women with COVID-19 pneumonia had elevated sFlt-1/PlGF and sFlt-1/ANG II ratios. These elevated ratios served as markers of poor prognosis, and specifically, the sFlt-1/ANG-II ratio was associated with an increased likelihood of developing severe pneumonia, requiring ICU admission, intubation, sepsis, and experiencing death. However, no association was observed between these ratios and complications related to PE [54].

In relation to the prognosis of COVID-19, studies conducted on non-pregnant populations have already identified an elevation in serum sFlt-1 levels among patients infected with SARS-CoV-2. These elevated levels have been recognized as an important marker of poor prognosis, particularly with regard to thrombotic events. This finding has prompted physicians to adopt more aggressive therapeutic approaches for patients with elevated sFlt-1 levels, aiming to enhance their chances of survival [55].

As mentioned previously, Mendoza et al. have reported that severe cases of COVID-19 infection can lead to the development of a PE-like syndrome. However, this condition can be differentiated from true PE by normal parameters observed in ultrasound, such as the uterine artery pulsatility index, and sFlt-1/PlGF ratio. In these cases, the abnormal sFlt-1/PlGF ratio does not indicate abnormal placentation but rather reflects the severity of the infection. Furthermore, markers return to normal in COVID-19-infected pregnant women before the delivery of the placenta, indicating the absence of placental dysfunction [20,56]. Importantly, they did not induce labor, and the patients recovered from the signs and symptoms of PE when they improved from severe pneumonia [22].

The current evidence supports the use and effectiveness of the sFlt-1/PlGF ratio in differentiating PE in relation to COVID-19. However, it is evident that relying solely on the ratio may not be sufficient for accurate differentiation. Analyzing the progression of the clinical condition and considering additional parameters, such as the pulsatility index of uterine arteries, proves to be more beneficial in this regard [22].

#### 2.2.5. Maternal Cardiovascular Dysfunction

COVID-19, an unprecedented pandemic, can have significant cardiovascular implications for the millions of individuals worldwide who survive the infection. SARS-CoV-2, this virus, has the ability to infect various cardiovascular tissues, including the heart, vascular tissues, circulating cells, and cells in the placenta (specifically trophoblast and cytotrophoblast cells) [57].

Current evidence supports the role of maternal cardiovascular dysfunction in the development of PE. This evidence is derived from studies showing an increased risk of both PE and fetal growth restriction in women with congenital heart disease compared to healthy pregnant women [58,59,60].

PE has been associated with a high prevalence of asymptomatic left ventricular dysfunction and myocardial injury, as compared to normotensive pregnant women [61]. Diastolic dysfunction and left ventricular remodeling are more pronounced in pregnant women with severe early-onset PE, and these can be detected before clinical manifestation and are associated with adverse pregnancy outcomes [62].

During the preclinical phase of PE, parameters such as the total peripheral resistance index, serum sFlt-1, and B-type natriuretic peptide (BNP) levels are elevated. A combined model incorporating these parameters successfully differentiated PE cases from controls with high precision (area under the curve: 0.96). These findings suggest that, before the onset of clinical PE, there is an increase of peripheral vasculopathy with the consequent increase in pressure in the ventricular walls, leading to higher BNP levels. These effects are most pronounced in early-onset PE, indicating more severe cardiac involvement [63].

In addition, several studies have described the deleterious effects of COVID-19 on the cardiovascular (CV) system. CV involvement is common in elderly patients with comorbidities, but acute myocarditis has also been described in adult patients with severe COVID-19 without previous comorbidities [64]. The exact mechanism underlying cardiac involvement in COVID-19 remains unclear. One possible mechanism is direct ACE-2-mediated myocardial involvement. Supporting this hypothesis is the finding that, during the Toronto SARS outbreak, SARS-CoV viral RNA was detected in 35% of the autopsied hearts. Other suggested mechanisms of COVID-19-related cardiac involvement include a cytokine storm mediated by an unbalanced response between T-helper cell subtypes and hypoxia-induced calcium excess, which would lead to apoptosis of cardiac myocytes [65].

In young, previously healthy non-pregnant women infected with SARS-CoV-2 and presenting with severe disease, associations have been observed with myocarditis, acute myocardial infarction, cardiomyopathies, arrhythmias, and venous thromboembolic events [66]. Although there are several mechanisms that can cause myocardial damage, myocarditis and systemic inflammation are the most common. Myocardial damage is considered a marker of mortality risk, more significant than age, previous CV disease, CV risk factors, or chronic lung disease. All patients in the study exhibited elevated levels of serological markers of myocardial damage, such as BPN and troponin [67,68].

To investigate the impact of COVID-19 on the hearts of pregnant women, Mercedes et al. conducted a retrospective study involving 15 previously healthy pregnant women with confirmed severe COVID-19. All patients were admitted to the ICU. Myocardial injury was determined based on significantly elevated levels of troponin in the serum (>0.4 ng/mL) (AA) and elevated Pro-B-type natriuretic peptide concentration. Additionally, structural and functional deterioration of the left ventricle was observed through transthoracic echocardiography. All patients exhibited left ventricular (LV) dysfunction, with an average LV ejection fraction of 37.67% ± 6.4 (normal value > 50%) and diffuse LV hypokinesis. These findings suggest a higher prevalence of COVID-19-induced systolic dysfunction in pregnant women compared to non-pregnant patients [69]. Critically ill pregnant women with CV dysfunction due to COVID-19 also have an increased maternal mortality rate (13.3%) [68,69].

In a similar vein to the previous study, Jusela et al. conducted research involving two cases of previously healthy pregnant women with COVID-19 who developed cardiomyopathy. These women exhibited CV dysfunction, characterized by a moderate reduction in LV ejection fraction (40–45%) and hypokinesia. It remains uncertain whether ventricular dysfunction in these patients is a result of direct viral effects on cardiac cells or multiple organ failure. However, diagnosing CV involvement is crucial to provide appropriate treatment and prevent complications [70].

Furthermore, other researchers, despite the limited population of pregnant women with coronavirus, discovered that, among pregnant women admitted to the ICU with hypertension or PE, there were elevated serum concentrations of Troponin I (cTn) and pro B-Type natriuretic peptide, as well as bradycardia, indicating myocardial injury [68,70].

Considering the evidence of cardiac involvement in both severe COVID-19 disease and PE, it has been proposed, and we agree with this perspective, that the increased incidence of PE observed during the pandemic in pregnant patients with severe COVID-19 may be linked to myocardial dysfunction that occurs during the disease. This dysfunction could be caused by massive systemic inflammation or direct damage to cardiomyocytes due to the entry of the virus into the cells. Pregnant women with severe COVID-19 disease may experience heart disease and myocarditis, which can lead to hypoperfusion and acquired ischemia in the placenta and possibly other organs. These conditions of hypoperfusion and ischemia disrupt the balance between the production of pro- and anti-angiogenic factors, ultimately contributing to the development of PE [71,72,73,74].

The hypothesis suggesting that myocardial injury associated with SARS-CoV-2 infection could potentially explain the occurrence of PE in infected pregnant women is intriguing and warrants further investigation.

Giogione also propose conducting non-invasive imaging studies and cardiac function tests in pregnant women infected with the virus to detect cardiac injury and provide appropriate treatment [75].

## 3. PE in Relation to SARS-CoV-2 Variants

Throughout the course of the COVID-19 pandemic, there have been multiple surges in infection rates primarily driven by successive variants of the SARS-CoV-2 virus, which have been designated with alphabetical labels in chronological sequence.

Observing how the prognosis of the association between PE and COVID-19 has evolved throughout the pandemic as new variants of the original virus have emerged and vaccination has taken place, we can note the following.

Most authors acknowledge three distinct periods of time in which each variant was predominant (excluding subvariants): the pre-Delta epoch, spanning from 17 May 2020 to 26 June 2021; the Delta epoch, from 27 June 2021 to 11 December 2021; and the Omicron epoch, from 1 December 2022 to 29 January 2022. The exact dates may vary depending on the location and the strategy employed to identify the variants [76]. Similar to individuals who are not pregnant, the prevalence of the Delta and Omicron variants was linked to a higher occurrence of SARS-CoV-2 infections during pregnancy, mainly observed in unvaccinated individuals. Compared with the pre-Delta epoch, periods of Delta and Omicron predominance were associated with increased infections. The dominance of the Delta variant was associated with more severe illness, whereas the Omicron variant was associated with milder illness, taking into account previous vaccination status. Among unvaccinated mothers with non-severe COVID-19, most cases of early neonatal SARS-CoV-2 infections were reported [27,76].

In a study where the objective was to assess the pregnancy outcomes of women who were not vaccinated and contracted SARS-CoV-2 during different waves of the pandemic, specifically the pre-Delta, Delta, and Omicron waves, they found that when contrasting the Omicron wave with the pre-Delta wave, it was noted that the maternal mortality rates remained comparable. Additionally, the severity of the disease and complications during pregnancy were also found to be similar, infection during the Omicron wave, when compared to the pre-Delta period, resulted in a comparable requirement for oxygen supplementation (in the manner it was defined). Furthermore, there were no disparities in the rates of maternal death or pre-eclampsia. In contrast, when comparing the Delta period to the pre-Delta wave, Delta was linked to an elevated need for oxygen supplementation (including ECMO) and a notable increase in the incidence of maternal death. Additionally, the rate of very preterm birth showed a significant rise during the Delta period [27].

The significance of vaccinating pregnant women cannot be overstated, especially during the Omicron wave when the effectiveness of vaccination against SARS-CoV-2 infection is reduced. It is crucial to effectively communicate the risks associated with the Omicron variant to prevent any negative impact on the vaccination rate among pregnant women. These women face a higher risk of experiencing adverse maternal and perinatal events due to COVID-19.

A systematic review and meta-analysis with a similar objective observed that Omicron infections are linked to less severe adverse outcomes for both mothers and newborns. However, it is important to note that maternal ICU admission, the need for respiratory support, and preterm birth can still occur with Omicron infections. Considering that Omicron is currently the dominant strain globally, with high transmission rates, it remains crucial to remain vigilant in safeguarding vulnerable populations such as mothers and infants. Subsequent statistics indicate that infection with the highly contagious Omicron variant during pregnancy resulted in milder disease outcomes compared to the preceding Delta variant. This is likely due to higher vaccination coverage, lower virulence of the Omicron variant, and acquired immunity from previous infections [28].

## 4. Conclusions

Pregnancy presents a risk factor for the development of severe COVID-19 in unvaccinated pregnant women, particularly those with underlying health conditions.

There are several shared pathophysiological mechanisms between COVID-19 and PE. Research studies have provided evidence of a connection between SARS-CoV-2 infection and PE, indicating a possible cause-and-effect relationship.

Given the abundance of published studies on COVID-19 in pregnant women, we aimed to review the literature specifically regarding the relationship between COVID-19 and PE. By describing their similarities and differences, our goal was to identify the key factors that contribute to the increased severity of both conditions when they occur simultaneously in a patient. Research studies have provided evidence of a connection between SARS-CoV-2 infection and PE, indicating a possible cause-and-effect relationship.

Pregnant women who experience symptomatic respiratory symptoms and/or more severe cases of coronavirus disease are at an increased risk of developing PE. While there is not enough evidence to establish whether the timing of infection during pregnancy influences the risk of PE, the most severe instances occur when both conditions coincide during the latter part of the second trimester and throughout the third trimester. This overlap leads to poorer obstetric and neonatal outcomes.

It is plausible that the complex relationship described between SARS-CoV-2 infection and acute severe CV dysfunction in COVID-19 patients may also occur during SARS-CoV-2 infection in pregnancy. Results from studies support this hypothesis, indicating a connection to massive systemic inflammation, as confirmed by elevated levels of troponin and pro-B natriuretic peptide, as well as LV dysfunction, both during and outside of pregnancy. Myocardial injury and subclinical CV dysfunction may contribute to acquired uteroplacental malperfusion and ischemia, potentially leading to PE in symptomatic and asymptomatic pregnant women.

The hypothesis suggesting that myocardial injury associated with SARS-CoV-2 infection could potentially explain the occurrence of PE in infected pregnant women provides the strongest evidence of association for our analysis and warrants further investigation. Strict follow-up should be conducted on patients and their children who experienced COVID-19 associated PE during pregnancy to observe long-term consequences, including long-term COVID-19 implications.

Given the ongoing global COVID-19 pandemic and the emergence of new variants, it is imperative to enhance research efforts focused on understanding the effects of these variants on the health of mothers and infants.

Vaccination against COVID-19, both in the general population and particularly in pregnant women, has dramatically reduced the severity of the disease.

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
