# Peer review of "SARS-CoV-2 Infection and Preeclampsia—How an Infection Can Help Us to Know More about an Obstetric Condition"

_viruses, 2023, doi:10.3390/v15071564_

Round 1

Reviewer 1 Report

In general, I find various types of errors throughout the text.

·         The abstract is too extended and does not meet the journal's requirements.

·         Only 3-5 keywords are required

·         There is no apparent difference between the abstract section and the beginning of the introduction.

·         It is necessary to standardize the use of terms such as COVID-19, SARS-CoV-2, and PE.

·         "Preeclampsia," "pre-eclampsia," and "PE" are used interchangeably, as well as "pro-inflammatory" and "proinflammatory."

·         The references do not comply with the format requested by Viruses.

·         Throughout the text, there is excessive space between words.

·         The typography format is not suitable for this journal. The individual contributions of each author, the corresponding acknowledgments, and the possible declarations of conflicts of interest related to the work are missing.

·         Check https://www.mdpi.com/journal/viruses/instructions

·         Figures are missing to help the reader better understand the concepts presented.

·         Reading is difficult because the content and ideas of the text do not flow coherently.

·         It is necessary to add specialized bibliography on PE and COVID-19.

The help of a native English speaker is needed to improve the grammar and style of the article.

Reviewer 2 Report

The manuscript is informative and explores the  links between SARS-CoV2 infection and preeclampsia.

There are few typos and some English grammar to improve.

Page 2: previous SARS and MERS  were "epidemic outbreaks" (not pandemics).

increased stroke "incidence" (not volume)

Page 3: it is "maternal vascular malperfusion" (not placental inssufficiency)

Page 9:4th paragraph: it is "Immunohistochemical analysis" (instead of histological examination)

However,  the style of the manuscript - a non-structured review0 will be better placed as a chapter in a book than as an original study in a Journal with high impact factor.

Many references are outdated, as they are from 2020 and 2021. Only a few are from early 2022. In addition, the incidence of preeclampsia and maternal/fetal mortality has significantly reduced during the Omicron wave, and the manuscript fell short to address this.

The authors need to correct some typos and grammar errors.

Round 2

Reviewer 1 Report

In general, I see the content of the writing well. However, the typography format is not the one indicated for Viruses. Let's hope that the editorial office will help in adjusting the design.

None